# Experimental Study on Carbon Fiber-Reinforced Composites Cutting with Nanosecond Laser

**DOI:** 10.3390/ma15196686

**Published:** 2022-09-27

**Authors:** Jihao Xu, Chenghu Jing, Junke Jiao, Shengyuan Sun, Liyuan Sheng, Yuanming Zhang, Hongbo Xia, Kun Zeng

**Affiliations:** 1School of Mechanical Engineering, Yangzhou University, Yangzhou 225009, China; 2PKU-HKUST ShenZhen-HongKong Institution, Shenzhen 518057, China; 3School of Mechanical and Vehicle Engineering, Linyi University, Linyi 276005, China; 4Yangzhou Hanjiang Yangzi Automobile Interior Decoration Co., Ltd., Yangzhou 225009, China

**Keywords:** carbon fiber reinforced composites, nanosecond laser, laser cutting, heat affected zone, surface roughness, hole taper

## Abstract

The carbon fiber-reinforced composite (CFRP) has the properties of a high specific strength, low density and excellent corrosion resistance; it has been widely used in aerospace and automobile lightweight manufacturing as an important material. To improve the CFRP cutting quality in the manufacturing process, a nanosecond laser with a wavelength of 532 nm was applied to cut holes with a 2-mm-thick CFRP plate by using laser rotational cutting technology. The influence of different parameters on the heat-affected zone, the cutting surface roughness and the hole taper was explored, and the cutting process parameters were optimized. With the optimized cutting parameters, the minimum value of the heat-affected zone, the cutting surface roughness and the hole taper can be obtained, which are 71.7 μm, 2.68 μm and 0.64°, respectively.

## 1. Introduction

Carbon fiber-reinforced composite (CFRP) has the advantages of a high specific strength, low density and light weight, and has become an important material for aerospace and automobile lightweight manufacturing [1,2]. Due to the great difference in physical properties between carbon fibers and the resin matrix in CFRP, problems such as delamination, fiber pull-out and tool wearing often occur when using traditional mechanical machining methods to cut CFRP. As a non-contact advanced processing technology, laser machining technology shows great potential in composite materials’ cutting [3]. However, due to the thermal machining process, the ablation of resin matrix materials easily occurs during laser cutting, leading to problems such as a heat-affected zone (HAZ), delamination and fiber pull-out, which seriously affect the processing quality of CFRP [4,5]. In order to improve the quality of CFRP laser cutting, a large number of investigations have been carried out on CFRP laser cutting in recent years.

Jiang cut a 5-mm-thick CFRP plate with lasers, and he found that the HAZ gradually increased with an increase of power and that the HAZ reached the minimum value when the laser power was 60 W. He also found that the HAZ decreased with an increase of the laser scanning speed and that it reached the minimum value when the scanning speed was 10 m/s [6]. Ye used a 532 nm wavelength laser to make holes in CFRP, and compared the effects of cross filling, parallel filling and rotary scanning on the machining quality. It was found that the cutting efficiency of laser rotary cutting was significantly higher than that of the other two methods and that the HAZ was smaller [7]. Song found that adding carbon black particles in CFRP can significantly improve the cutting effect and reduce the occurrence of cracks and delamination [8]. Hua used a 500 W millisecond pulse laser to cut CFRP underwater. It was found that the HAZ was greatly reduced when compared with that cutting in air [9]. Zhang used lasers with different wavelengths to make holes in CFRP plates. It was found that the shorter the laser wavelength was, the higher the cutting surface quality and the smaller the HAZ that could be obtained [10]. Bluemel found that a continuous wave laser has a higher cutting speed than a pulsed laser but that the nanosecond laser has a narrower slit and higher tensile strength [11]. Negarestani found that when using mixed gas containing oxygen and inert gas as the auxiliary gas in CFRP cutting, the HAZ could be greatly reduced [12]. Li found that the cutting speed is an important factor affecting the HAZ, while the laser processing parameters have a limited influence on the surface strain distribution during the tensile load [13]. Lau found that the cutting direction of the laser affected the width of the HAZ and the cutting depth [14]. Jose found that cutting CFRP with a high repetition rate resulted in a very short cooling time and a large HAZ. To this end, a higher pulse duration and lower repetition frequency could reduce the HAZ [15].

As mentioned above, parameters such as the wavelength, the frequency, the laser mode (pulsed or continuous), the laser power and the cutting speed will influence the cutting quality of CFRP. Compared with the continuous laser, the short-pulsed laser can reduce the cutting HAZ and improve the cutting accuracy. Compared with the infrared laser, a laser with a short wavelength has a smaller heat input and smaller taper when cutting CFRP. Therefore, using a short-wavelength and short-pulsed laser to cut composites has become a trend to improve the cutting quality of CFRP. However, relations between the cutting quality, such as HAZ, surface roughness and the taper of the cutting hole, and the pulsed laser parameters, such as laser power and laser scanning speed, are not clear. It necessary to investigate them and make conclusions on the technical process, which is helpful in getting a high CFRP cutting quality with short-pulsed lasers in industry applications. To this end, in this paper, a nanosecond laser with a wavelength of 532 nm was applied to cut holes in a 2-mm-thick CFRP plate. Twenty-five groups of experiments with different cutting parameters were designed. The HAZ and the cutting surface roughness was measured by using SEM and a laser confocal microscope, and the taper of the cutting hole was calculated. The influence of different parameters (the laser power and the laser scanning speed) on the HAZ, the surface roughness and the hole taper was explored, and the cutting parameters were optimized, with the aim of laying a technological foundation for high-quality CFRP laser cutting.

## 2. Materials and Methods

The size of CFRP is 2 mm × 100 mm × 100 mm, and the volume contents of the carbon fibers and the epoxy resin are 68% and 32%, respectively. The fibers in the CFRP are continuous fiber, and the diameter of the fiber is about 5 μm. The CFRP was produced by Ningbo Institute of Materials technology and engineering. Additionally, the thermos-physical parameters of CFRP are shown in Table 1 [16].

A nanosecond laser is used to cut CFRP in this work. The laser wavelength is 532 nm, the pulse duration is 150 ns, the maximum power is 35 W, and the repetition frequency can be changed in the range of 50 KHz to 200 kHz. The parameters that can be set by the machining system include the scanning speed, the laser power and the repetition frequency of the nanosecond laser. The hole-cutting path and the cutting speed can be set by the scanning galvanometer. Figure 1a is a diagram of the nanosecond laser machining system. In the previous study, it was found that the cutting quality and efficiency of laser rotary cutting were higher than those of cross filling and parallel filling [16]. Therefore, the laser rotary cutting technique was used to cut CFRP in this work (Figure 1b). Furthermore, the spacing *d* was set at 0.1 mm and the radius *r* was set at 0.1 mm, and the radius of the cutting hole *R* was 2.5 mm, which was optimized in our previous work [16].

In our previous research, it was found that the amount of energy (*E*) absorbed per unit area per unit time was determined by the laser power *P*, the laser scanning speed *V* and the laser spot diameter *D*, which is shown in Equation (1) [17]:(1)$$E=A\times \frac{P}{V\times D^{2}}$$
where *A* is the constant coefficient, which is determined by the laser absorption coefficient of the CFRTS.

In order to study the influence of different scanning speeds *V* and laser powers *P* on the CFRP machining quality, an orthogonal experiment was designed in this paper. The pulse duration was 150 ns, the repetition frequency was 60 KHZ, the laser power *P* was set at 21 W, 24.5 W, 28 W, 31.5 W and 35 W, and the cutting speed *V* was set at 800 mm/s, 900 mm/s, 1000 mm/s, 1100 mm/s and 1200 mm/s, respectively. A total of 25 groups of experiments were conducted, as shown in Table 2. During laser processing, nitrogen with a purity of 99.99% is used as the protective gas, and the nitrogen flow rate is about 8 L/min. Nitrogen can play the role of cooling the CFRP plate, reducing the generation of the HAZ, and blowing away the slag to improve the cutting efficiency. In order to observe the roughness of the cutting surface under different parameters, the CFRP plate was cut into 5-mm-wide strips. After cutting, the machining debris were removed by ultrasonic cleaning. During the measurement, three different positions on the cutting surface were measured, and the average value was calculated. The HAZ was measured by using a microscope, as shown in Figure 2a. From this figure, it can be seen that there is a region around the hole whose color is different from the CFRP matrix, and the width of this region is defined as the HAZ. The surface roughness of the CFRP cutting profile was detected by using the laser confocal microscope, as shown in Figure 2b. In this work, the size of the scanned area for the roughness measurement was 1054 μm × 1406 μm, and the average of *Ra* was chosen as the value of the surface roughness. After laser cutting, the microstructure of the cutting profile, such as fiber fractures and interlaminar tears, was detected by using SEM.

The laser will produce a certain divergence angle during the actual operation. Therefore, there will be a certain taper during laser processing, as shown in Figure 3. In this experiment, the average value and standard deviation of the three tapers are calculated by three measurements at the entrance and exit of the beam. The taper *θ* can be calculated by the following formula:(2)$$\theta =\arctan\frac{D_{1}-D_{2}}{2h}\times \frac{180{}^{\circ}}{\pi }$$
where *D*_1_ is the beam entrance size, *D*_2_ is the beam exit size, and *h* is the thickness of the CFRP plate.

## 3. Results and Discussions

### 3.1. Cutting Holes in CFRP Plate with Nanosecond Lasers

The process parameters of CFRP laser rotary cutting include the rotation diameter, the scanning space and the scanning speed. The focal length compensation speed of the z-axis is directly related to the above parameters. The focal length compensation speed should be matched with the material removal speed. The material removal ability with the negative defocus will become weak, or there will even be no removal ability. If the focal length compensation speed is too slow, it will affect the removal speed. Appropriate mathematical models and multiple experiments should be established to determine the appropriate calculation formula for the focal length compensation speed [7]. Finally, it is determined that the focal length of the 2-mm-thick plate is 0.5 mm per minute and that the time needed to punch through the hole is about 4 min. The holes of Sample 1-1 to Sample 5-5, cut with the parameters shown in Table 2, are shown in Figure 4a, and the setting diameter of the holes is 5 mm. As shown in Figure 4b,c, the inlet diameter is about 5.06 mm and the outlet diameter is about 4.96 mm when the laser power is 28 W, and the scanning speed is 1200 mm/s (Sample 5-3). The hole taper can be calculated by using the formula (2), as mentioned above, which is about 0.71°.

To clarify the HAZ in the CFRP matrix during the laser cutting process, the HAZ around the hole and the microstructure of the cutting profile were detected. Figure 5 shows the HAZ and the microstructure on the CFRP hole cutting surface for Sample 5-3. From Figure 5b, it can be seen that there is a circa 140-μm-wide HAZ around the inlet edge. With the laser interaction, the CFRP material is removed from the hole inlet during the cutting process, and the plasma produced during the cutting process goes away from the hole. Because of the high plasma temperature, the hole surface is heated and ablated, and a HAZ is generated. From Figure 5b, it can be seen that there are serrations around the hole inlet, which is caused by the excessive rotation space *d* of the rotating laser beam, as shown in Figure 1b. These serrations can be avoided by reducing the rotation space *d*. After cutting holes with the nanosecond laser, the CFRP hole is cut along the center line by lasers, and the hole profile is shown in Figure 5c,d. From this figure, it can be seen that the CFRP surface for nanosecond laser cutting is smooth, and there is no delamination and fiber pull-out on the cutting surface.

The CFRP materials are vaporized due to the high energy of the pulsed nanosecond laser, and hence there is no mechanical force and impact force during the cutting process. There is no carbon fiber drawing and tearing compared with the mechanical cutting method. The HAZ can be controlled well, the cutting surface has a low roughness, and the taper of the cutting hole is about 0.71°. With this CFRP laser cutting technique and the parameters of Sample 5-3, holes with different shapes such as triangle and square can be drilled by changing the laser beam path (Figure 6). It can be seen that the cutting profile is clean and that there are no burrs and tears on the cutting surface. 

### 3.2. CFRP Cutting Quality with Different Laser Parameters

The HAZ is the key factor affecting the cutting quality of CFRP. Figure 7 shows the HAZ of the hole edge under different powers and cutting speeds. It can be seen from the figure that when the laser power is 21 W, the HAZ first decreases and then gradually increases with the increasing of the laser scanning speed. When the scanning speed is 900 mm/s, the HAZ decreases first and then increases when increasing the laser power, and the minimum HAZ value of about 71.7 μm can be obtained when the laser power is 28 W. For other scanning speeds such as 800 mm/s, 1000 mm/s, 1100 mm/s and 1200 mm/s, nearly the same trend can be obtained, as shown in Figure 7. 

The minimum value is 122 μm at a speed of 1000 mm/s (Sample 3-1), and the maximum value is 170.3 μm at a speed of 1200 mm/s (Sample 5-1). When the laser power is 24.5 W, the HAZ increases first and then decreases when increasing the scanning speed, and the minimum value is 71.9 μm at a speed of 1200 mm/s (Sample 5-2). When the laser power is 28 W, the minimum value of HAZ is 71.7 μm at a scanning speed of 900 mm/s (Sample 2-3). In the experiment, the rotation radius and spacing of laser cutting can be reduced to reduce the serration of cutting. Increasing the flow of nitrogen shielding gas to cool the CFRP plate in a timely way could reduce the HAZ.

The surface roughness was measured by using a laser confocal microscope, and the roughness was kept within 12 um for these 25 cutting samples, as shown in Figure 8. From this figure, it can be seen that for a specific laser scanning speed, the cutting surface roughness increases first and then decreases when increasing the laser power. The minimum cutting profile roughness can be obtained when the laser power is 35 W and the scanning speed is 900 mm/s (Sample 2-5), which is about 2.68 μm. From Figure 8, it also can be seen that there is little change in the surface roughness with the change of the laser scanning speed when the laser power is 35 W, and the minimum value can be obtained when the speed is 900 mm/s. This means that a cutting profile with a low roughness can be obtained when the laser power is 35 W.

On the other hand, there is no obvious law between the roughness and the laser scanning speed. However, the minimum value roughness can be obtained under different laser scanning speeds, such as a minimum roughness of 3.08 μm when the laser power is 21 W and the scanning speed is 800 mm/s (Sample 1-1), 3.22 μm when the power is 35 W and the scanning speed is 1000 mm/s (Sample 3-5), 3.66 um when the power is 35 W and the scanning speed is 1100 mm/s (Sample 4-5), and 5.06 μm when the laser power is 35 W and the scanning speed is 1200 mm/s (Sample 5-5). 

The hole taper can be calculated by measuring the diameter of the inlet and outlet of the hole, as shown in formula (1). From Figure 9, it can be seen that the hole taper can be controlled within 1.6° in the experiment. For a specific laser scanning speed, the hole taper increases first and then decreases when increasing the laser power. When the scanning speeds are 800 mm/s and 1200 mm/s, it can be seen that the hole taper decreases first and then increases when increasing the laser power, and the minimum values, which are about 0.75° and 0.71°, respectively, can be obtained when the laser power is 28 W. The minimum taper can be obtained when the laser power is 24.5 W and the scanning speed is 1100 mm/s (Sample 3-2), and it is about 0.64°. 

From this figure, it also can be seen that the difference between the minimum hole taper under different powers is small, at 0.84°, 0.64°, 0.71°, 0.88° and 1.20°, when the laser power is 21 W (Sample 4-1), 24.5 W (Sample 4-2), 28 W (Sample 5-3), 31.5 W (Sample 5-4) and 35 W (Sample 5-5), respectively. To this end, it can be concluded that the hole taper is less affected by laser power and that the taper is relatively stable. In later experiments, the focal length of the laser can be further controlled to keep the focal length on the cutting surface, which can further reduce the taper.

## 4. Conclusions

In this paper, a 532-nm-wavelength nanosecond laser was used to cut a 2-mm-thick CFRP plate. The effects of different laser powers and cutting speeds on the HAZ, surface roughness and taper are investigated. The following conclusions can be obtained. 

(1)Laser rotational cutting technique can improve the efficiency and quality of cutting, but there will be small serrations, which can reduce the radius and spacing of laser rotary cutting and reduce the serration of cutting.(2)Together, the laser power and the laser scanning speed affect the HAZ. Additionally, the amount of energy absorbed per unit area per unit time is determined by both of them. When the power is 28 W and the scanning speed is 900 mm/s, the minimum value of the heat-affected zone can be obtained, and it is about 71.7 μm.(3)The cutting surface roughness increases first and then decreases when increasing the laser power for a specific laser scanning speed. The minimum value is 2.68 μm when the scanning speed is 900 mm/s and the laser power is 35 W.(4)The hole taper increases first and then decreases when increasing the laser power for a specific laser scanning speed. The minimum taper is 0.64° when the laser power is 24.5 W and the scanning speed is 1100 mm/s.

## Figures and Tables

**Figure 1 materials-15-06686-f001:**
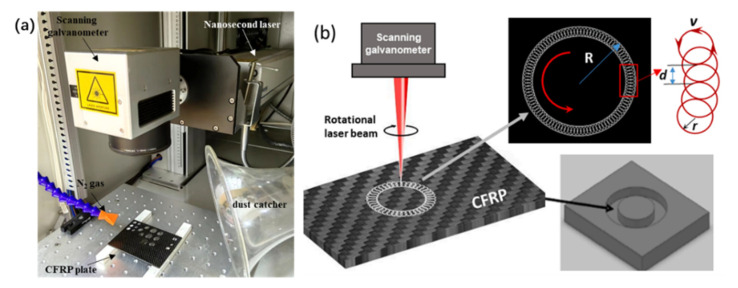
(**a**) The nanosecond laser machining system; (**b**) cutting CFRP with the laser rotary cutting technique [16].

**Figure 2 materials-15-06686-f002:**
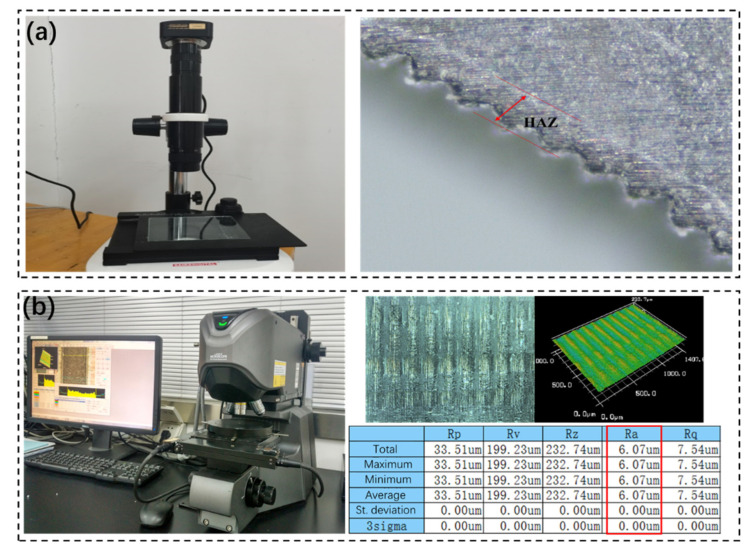
(**a**) Measuring the HAZ around the cutting hole with the microscope; (**b**) measuring the surface roughness with the laser confocal microscope.

**Figure 3 materials-15-06686-f003:**
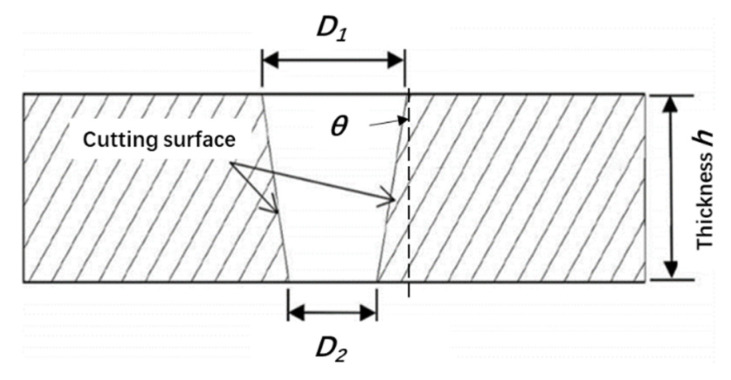
Hole taper during CFRP laser cutting.

**Figure 4 materials-15-06686-f004:**
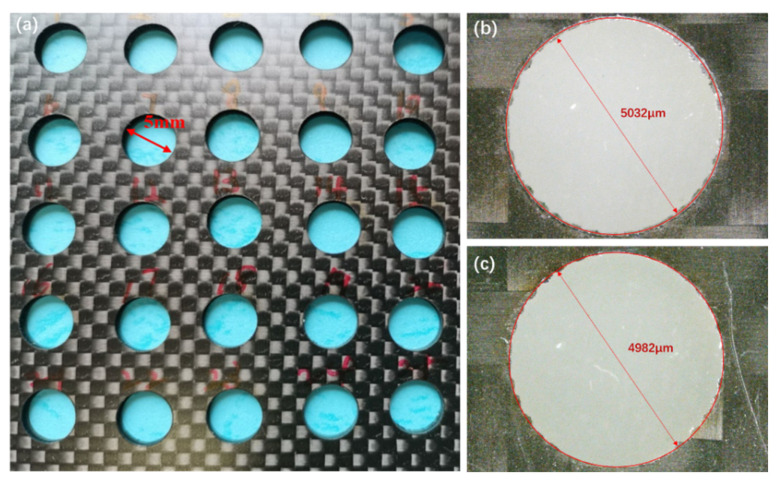
(**a**) Cutting holes in CFRP plate with nanosecond lasers for Sample 1-1 to Sample 5-5; (**b**) inlet of the CFRP laser cutting hole of sample 5-3; (**c**) outlet of the CFRP laser cutting hole of sample 5-3.

**Figure 5 materials-15-06686-f005:**
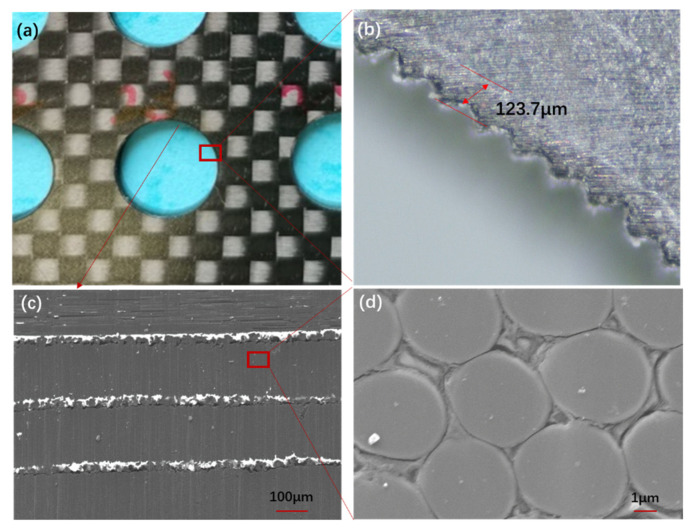
(**a**) Cutting hole in CFRP plate; (**b**) HAZ around the cutting hole; (**c**) cutting profile; (**d**) microstructure of the cutting profile.

**Figure 6 materials-15-06686-f006:**
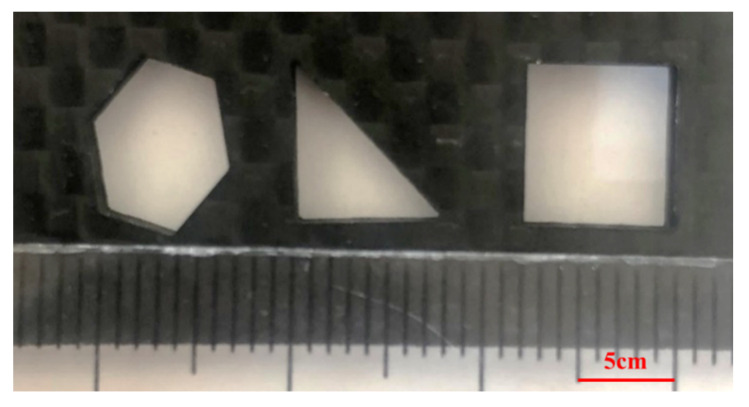
CFRP laser cutting holes with different shapes.

**Figure 7 materials-15-06686-f007:**
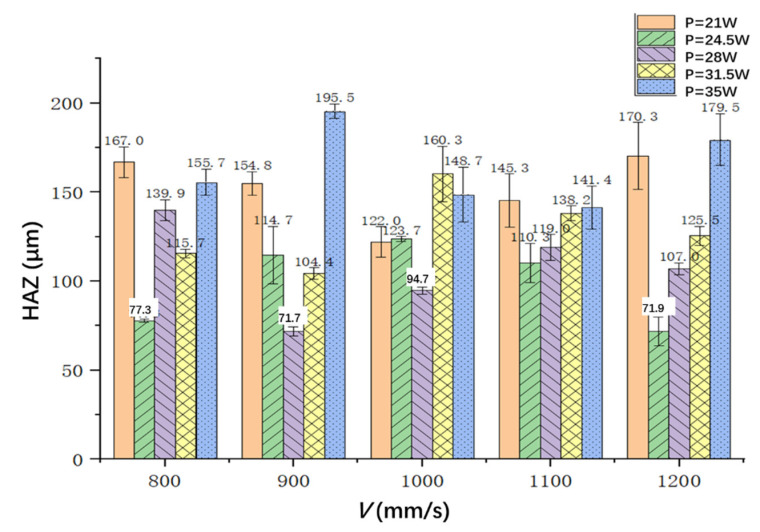
The HAZ with different laser powers and scanning speeds.

**Figure 8 materials-15-06686-f008:**
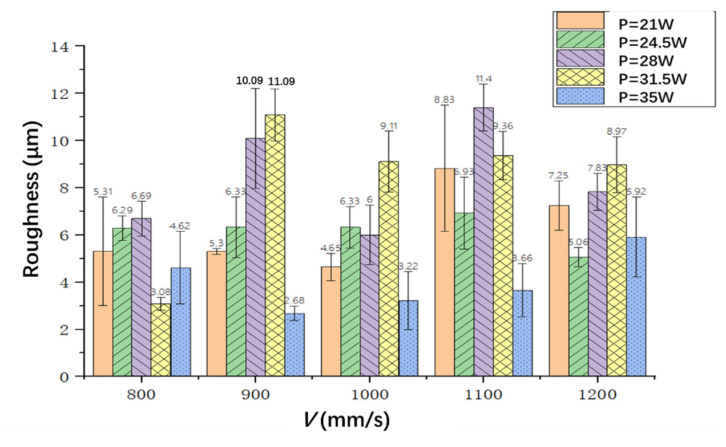
The surface roughness with different laser powers and scanning speeds.

**Figure 9 materials-15-06686-f009:**
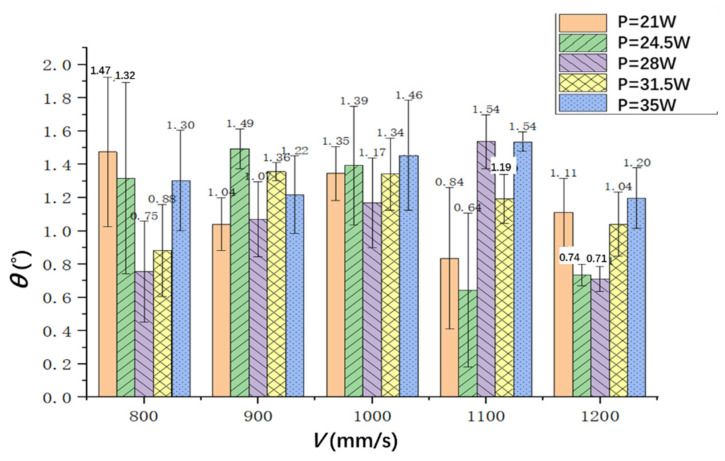
The hole taper with different laser powers and scanning speeds.

**Table 1 materials-15-06686-t001:** The thermos-physical parameters of CFRP.

Material	Thermal Conductivity (W⋅m^−1^ °C^−1^)	Specific Heat (J⋅kg^−1^⋅°C^−1^)	Vaporization Temperature (°C)	Density (g/cm^3^)
Epoxy resin	0.2	550	400	1.2
Carbon fibers	10.5	795.5	3300	1.76

**Table 2 materials-15-06686-t002:** Experimental design of CFRP laser cutting.

Parameter	21 W	24.5 W	28 W	31.5 W	35 W
800 mm/s	Sample 1-1	Sample 1-2	Sample 1-3	Sample 1-4	Sample 1-5
900 mm/s	Sample 2-1	Sample 2-2	Sample 2-3	Sample 2-4	Sample 2-5
1000 mm/s	Sample 3-1	Sample 3-2	Sample 3-3	Sample 3-4	Sample 3-5
1100 mm/s	Sample 4-1	Sample 4-2	Sample 4-3	Sample 4-4	Sample 4-5
1200 mm/s	Sample 5-1	Sample 5-2	Sample 5-3	Sample 5-4	Sample 5-5

## Data Availability

Data sharing not applicable.

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
