# Peer review of "Experimental Study on Carbon Fiber-Reinforced Composites Cutting with Nanosecond Laser"

_materials, 2022, doi:10.3390/ma15196686_

Round 1

Reviewer 1 Report

The manuscript presents an experimental investigation on carbon fiber reinforced composites cutting with nanosecond laser. a 532nm wavelength nanosecond laser was used to cut a 2mm thick CFRP plate. The effects of different laser power and cutting speed on the HAZ, surface roughness and taper are investigated in this study. The paper is focusing on an interesting topic however it is difficult to read and some complementary explanations are highly needed in the revised version. The manuscript would benefit, once revised content-wise. The following points should be considered by the authors in the revised manuscript before acceptance for publication.

·         Some symbols don’t have description in the paper. It would better if a NOTATION added at the interval of the KEYWORDS and the INTRODUCTION, to present all the symbols in this paper.

·         The introduction section has documented "well" the recent and relevant literature. However, the new contribution of this work is not clear. In other words, what is missing and has not been investigated in the previous literature which is addressed in this manuscript. Could the authors comment.

·         The last paragraph of the introduction should be really improved. Indeed, the authors must rephrase this part in terms of experimental findings and to emphasis the paper’s structuration.

·         It’s very important that the authors add more results about the effect of the frequency, the laser modes, thickness of specimens?

·         The relation between the laser power vs scanning speed, and laser power vs toughness is not clear. I ask the authors to be constructive and give more details.

·         It is a good report, but which are the lessons learnt? The authors have to clarify this otherwise the paper cannot be accepted

·         The authors must improve the quality of all figures (300dpi)

Reviewer 2 Report

The authors present an Experimental study on carbon fiber reinforced composites cutting with nanosecond laser. They show results by varying parameters during the cutting process. I consider that the work could be accepted carrying out major corrections:

Section 1. Introduction

·       authors should improve the state of the art.

Section 2. Materials and Methods

·       The authors present some thermo-physical properties of CFRPs. They should comment on whether these properties were measured or how they were obtained.

·       The authors mention the present composition of the materials that form the CFRPs.

·       They must mention if the samples were purchased, or they prepared them.

·       The authors should present some characteristics of carbon fibers ie length and diameter.

·       In line 83 -84 the authors show the characteristics of the laser used to carry out the cuts of the samples. “The laser wavelength is 532nm, the pulse duration is 15ns, the maximum power is 35W”

However, in line 95-96. They comment “The pulse duration was 150ns, the repetition frequency was 60KHZ, the laser power was set as 21W”.

Were the authors able to clarify the duration of the pulse.

Section. 3 Results and discussions

·       The authors show many samples obtained from the cuts carried out. However, they should make a better discussion of their results.

·       The authors should mention the conditions under which the roughness measurements were carried out. Some norms or standard was used to perform the measurements.

·       What was the size of the scanned area for the roughness measurement?

·       Figure 7 shows the HAZ with different laser power and scanning speed. The variation of speed and power are shown in the graphs. The HAZ does not show a clear speed effect behavior at low speeds. However, for high speeds of 1000 and 1100 the HAZ increases as power increases. Authors should further discuss these results and support them with references.

·       Fig. 8. The roughness with different laser power and scanning speed. The presented results do not show a behavior based on the laser power and scanning speeds. The authors should do a better discussion of the results obtained.

·       Do the lines shown on the bars in figures 7 and 8 correspond to the standard deviation of the measurements?

Reviewer 3 Report

The author has conducted a comprehensive study of cutting CFRP by a nanosecond laser with the wavelength of 532 nm. The effects of experimental parameters on CFRP are discussed. The method of SEM characterized the structural development of materials. However, the results are not fully defined and deeply discussed. Also, some parts of the information are missing. Thus it is not qualified to publish on Materials.

1) P2L76-78: please add the vendor information for this CFRP material;

2) The unit in Table 1 needs to be corrected (e.g., m-1, kg-1, cm3…);

3) P3L83-84: What’s the laser source? CO2?

4) In Fig. 2 (a), what’s the hole size?

5) The scale bar in Fig. 6 is lacking units;

6) Where is the characterization method description? How to measure the roughness in this paper?

Round 2

Reviewer 1 Report

The manuscript can be accepted.

Reviewer 2 Report

The manuscript has been proofread as requested. I consider that I could be accepted as it is presented.

Reviewer 3 Report

This article has been revised based on the reviewer's suggestions, and its quality can be accepted now.